# Reactions of Trifluorotriacetic Acid Lactone and Hexafluorodehydroacetic Acid with Amines: Synthesis of Trifluoromethylated 4-Pyridones and Aminoenones

**DOI:** 10.3390/molecules27207098

**Published:** 2022-10-20

**Authors:** Vladislav V. Fedin, Sergey A. Usachev, Dmitrii L. Obydennov, Vyacheslav Y. Sosnovskikh

**Affiliations:** Institute of Natural Sciences and Mathematics, Ural Federal University, 51 Lenina Ave., 620000 Ekaterinburg, Russia

**Keywords:** 4-hydroxy-2-pyrone, 4-pyridone, aminoenone, trifluoromethylated heterocycles, regioselective reactions

## Abstract

Dehydroacetic acid and triacetic acid lactone are known to be versatile substrates for the synthesis of a variety of azaheterocycles. However, their fluorinated analogs were poorly described in the literature. In the present work, we have investigated reactions of trifluorotriacetic acid lactone and hexafluorodehydroacetic acid with primary amines, phenylenediamine, and phenylhydrazine. While hexafluorodehydroacetic acid reacted the same way as non-fluorinated analog giving 2,6-bis(trifluoromethyl)-4-pyridones, trifluorotriacetic acid lactone had different regioselectivity of nucleophilic attack compared to the parent structure, and corresponding 3-amino-6,6,6-trifluoro-5-oxohex-3-eneamides were formed as the products. In the case of binucleophiles, further cyclization took place, forming corresponding benzodiazepine and pyrazoles. The obtained 2,6-bis(trifluoromethyl)-4-pyridones were able to react with active methylene compounds giving fluorinated merocyanine dyes.

## 1. Introduction

Oxygen heterocycles are common precursors in organic synthesis and have a special place in the synthesis of important nitrogen heterocycles by various methodologies. Among others, pyrones attract significant interest both due to a variety of chemical properties and a wide distribution in nature [1,2,3,4]. Particular attention is paid to 4-hydroxy-2-pyrones, namely, triacetic acid lactone (TAL), dehydroacetic acid (DHA, Figure 1), and their derivatives, which belong to polyketides and occur widely in living organisms [5,6,7,8]. The availability of these compounds and the possibility of their biochemical synthesis contributed to their extensive research and to the development of a wide range of options for their modification, considering them as platform compounds [9,10,11,12].

On the other hand, the introduction of fluorine into the structure of molecules often leads to a modification of their chemical properties [13] and is a common strategy in drug design [14,15,16,17,18], pesticides synthesis [19,20], and other areas [21,22,23,24]. However, there are very limited data on fluorine-containing analogs of TAL and DHA in the literature, which practically do not cover their properties. The only synthesis of 4-hydroxy-3-trifluoroacetyl-6-trifluoromethyl-2*H*-pyran-2-one (hexafluorodehydroacetic acid, DHA-*f*_6_, Figure 1) was performed by the heating of trifluoroacetoacetic ester in the presence of P_2_O_5_ and proceeds through formation and dimerization of trifluoroacetylketene [25]. Hydrolysis of DHA-*f*_6_ in aqueous NaHCO_3_ leads to 4-hydroxy-6-trifluoromethyl-2*H*-pyran-2-one (trifluorotriacetic acid lactone, TAL-*f*_3_, Figure 1) [25]. The second route for the synthesis of TAL-*f*_3_ presented in the literature comprises cyclization of the corresponding trifluorodioxocaproic acid with acetic anhydride [26]. Among the properties of TAL-*f*_3_, only modification of the 4-OH group is known, which was methylated with Me_2_SO_4_ [27] and triflated with Tf_2_O [26]. The ring-opening of the O-methyl derivative on treatment with magnesium methylate gave methyl 3-methoxy-5-oxo-6,6,6-trifluoro-5-hexenoate [27] and the 4-OTf derivative reacted with 4-(methylthio)phenylboronic acid to give a Suzuki coupling product [26].

Thus, compounds that are attractive in terms of reactivity and properties of possible products turned out to be completely unexplored, and in this work, we have studied the interaction of DHA-*f*_6_ and TAL-*f*_3_ with primary amines as typical nucleophiles.

## 2. Results and Discussion

To begin with, we reproduced the synthesis described by German et al. (Figure 1) [25], as it gives access both to DHA-*f*_6_ and TAL-*f*_3_. The results of the first step were quite inconsistent, and the yield ranged from 0% to 75%, presumably due to heating with a flame burner, which is hard to control. Nevertheless, the product was obtained in a satisfactory quantity for the further research. It should be noted that no variation in reaction conditions has improved the yield. Thus, the lowering of P_2_O_5_ loading or increasing reaction time resulted in the formation of complex mixtures, either of open-chain condensation products and trimerization ones characterized by the ester methylene quartets in ^1^H NMR spectra or triads of equal intensity in ^19^F NMR spectra correspondingly. The hydrolysis of DHA-*f*_6_ was more reproducible; however, we could never achieve the literature yield of 95% and had 40% on average (Figure 1). Apart from NaHCO_3_, we have used NaOH and Na_2_CO_3_ as a base, but the yields were slightly lower. Adjusting pH to 4–5 during hydrolysis as the original procedure claims was quite difficult because it dropped down in time due to trifluoroacetic acid formation, so 2.0 equiv. of NaHCO_3_ was used instead. Moreover, crystallization of TAL-*f*_3_ did not occur until acidification to pH 0–1. We tried, as well, to carry out the detrifluoroacetylation of DHA-*f*_6_ by boiling it in water or by treatment with 93% H_2_SO_4_ or 70% HClO_4_ at room temperature, but the conversion was negligible.

At the first step, we studied the reaction of DHA-*f*_6_ with aniline. There are three main consecutive products that can be expected based on the literature data (Figure 2) [28,29]. While a formation of Schiff bases usually proceeds readily in acidic media, no conversion was observed when we attempted the synthesis of intermediate **A** in aqueous HCl with 1.5 equiv. of PhNH_2_ (Table 1, entry 1). When EtOH was used as a solvent, spontaneous decarboxylation took place and *N*-phenylpyridone **3a** was isolated in 60% yield (Table 1, entry 2). Formation of intermediate **B** was also observed and will be discussed later. Increasing the amount of PhNH_2_ and carrying out the reaction in a less polar solvent or without it as well as raising the temperature did not improve the yield (Table 1, entries 3–10).

Exploring the scope of the synthesis of 4-pyridones **3** from DHA-*f*_6_, we found that the reaction rates and the yields correlate with nucleophilicity of amine. Thus, aniline derivatives bearing π-donor substituents (MeO, F) at the 4-position react considerably faster, whereas those bearing strong acceptor substituents (Ac, CF_3_) at the 3-position or a weak acceptor substituent (Br) at the 4-position react at a comparable rate but give slightly lower yields (Figure 3, Table 2). Noteworthy, even extremely weakly nucleophilic 4-nitroaniline was able to react although the conversion was low even after 6 days, and pyridone **3k** was formed in only 8% yield. The characteristic signals of pyridones **3** in the NMR spectra acquired in CDCl_3_ are the singlet of vinylic CH with double intensity at 6.95–7.02 ppm (^1^H NMR) and the quartets of symmetric carbons at 119.2–119.4 ppm (CF_3_), 140.8–141.6 ppm (C-2 and C-6), and 119.0–119.6 ppm (C-3 and C-5).

The existence of an *ortho*-substituent did not affect the initial two steps of the reaction sequence and CO_2_ evolution occurred during first 1–2 h; however, the cyclization of bisenamines **B** depends on electron properties of the substituent. In the case of more reactive *o*-toluidine and 2,5-xylidine, the overall duration was 24 h, whereas 2-chloro- and 2,5-difluoroaniline required twice as much time (Table 2). It allowed us to isolate and characterize the intermediate **B′** (Figure 3), although in the mixture with starting 2-chloroaniline, which did not separate by the column chromatography. Its structure was confirmed by very distinguishable singlets of symmetric groups at 5.78 ppm (vinylic CH) and 11.49 ppm (enaminone NH) in ^1^H NMR and at 97.8 ppm in ^19^F NMR spectra in CDCl_3_. The formation of adducts with the structure of **B** was observed by TLC in all the cases, but for most of them the simultaneous cyclization to pyridones **3** occurred, so their isolation was impractical.

Surprisingly, aliphatic amines such as butylamine, *N*,*N*-dimethylethylenediamine and benzylamine did not react with DHA-*f*_6_ under the conditions found, even after several weeks. It can be attributed to their much higher basicity and fixation of the substrate in an anionic form that prevents further nucleophilic attack. This assumption was supported by the formation of the salt **4** (Figure 3) in the reaction with 2-aminopyridine, which precipitated in nearly quantitative yield and remained unchanged after 8 days at 60 °C in EtOH. The signals in the ^19^F NMR spectrum of compound **4** in DMSO*-d*_6_ almost exactly match the signals of DHA-*f*_6_ (91.0 and 89.8 ppm compared to 91.1 and 89.6 ppm, respectively) and the singlet of pyrone CH in the ^1^H NMR spectrum is shifted upfield by 0.09 ppm (6.15 compared to 6.24 ppm) in agreement with its anionic character.

Reactivity of 4-pyridones **3** is greatly affected by strongly acceptor CF_3_-groups. Thus, electrophilic bromination of **3a** with NBS proceeded poorly compared to the non-fluorinated analog [30] and no distinct product was obtained. On the contrary, Knoevenagel condensation did well with malononitrile, barbituric acid, and indane-1,3-dione in acetic anhydride, leading to fluorinated merocyanine dyes **5** (Figure 4, Table 3). This reveals that the electron deficiency of a heterocycle plays an important role for this reaction, as 1-aryl-2,6-dimethyl-4-pyridones are less active [31,32] compared both to the CF_3_-substituted pyridone **3a** and to the oxygen analog, 2,6-dimethyl-4-pyrone [33]. The time required for the reaction completion is in accordance with the acidity of an active methylene compound, so one can assume that deprotonation is a rate-limiting step, and a high activity of a substrate is necessary to prevent self-condensation of CH_2_X_2_.

The prepared compounds, **5a**–**c**, are found to be yellow solids and exhibit an absorbance major maximum at 382–437 nm in the visible region of the spectra. For barbituric derivative **5b**, the high molar extinction coefficient (91406 M^−1^·cm^−1^ at 412 nm) is observed and probably indicates the intramolecular charge-transfer described by the aromatic resonance form **5′** (Figure 2). For dihydropyridine **5c** bearing the indanedione moiety, additional intensive absorption maxima are observed at 246 and 224 nm. An interesting feature in NMR spectra of compounds **5** is a large difference compared to each other in chemical shifts of CH-groups in the dihydropyridine fragment attributed to the magnetic anisotropy of carbonyl and cyano groups, which have a closer proximity in derivative **5b** (*δ*_H3_ = 9.77 ppm, *δ*_C3_ = 116.8 ppm) than in **5c** (*δ*_H3_ = 9.10 ppm, *δ*_C3_ = 113.4 ppm) and in **5a** (*δ*_H3_ = 7.26 ppm, *δ*_C3_ = 112.7 ppm). Another possible explanation for the difference is a greater contribution of the betaine resonance form **5′** (Figure 2) for the adduct with barbituric acid, but this seems to have a smaller effect as the chemical shifts of the rest of carbon atoms diverge much less (Appendix A).

Moving on to the investigation of trifluorotriacetic acid lactone (**2**) properties, we focused on our previous work on the study of triacetic acid lactone (**6**) [34] and 2-cyano-6-(trifluoromethyl)-4*H*-pyran-4-one (**7**) [35], which gave carbamoylated enaminones **8** (Figure 5), which proved to be a versatile building blocks for nitrogen heterocycles. In both cases, the substrates were attacked with two equivalents of amine at positions 2 and 6.

We started the optimization of the reaction conditions of TAL-*f*_3_ (**2**) with aniline in EtOH using various amount of amine (Figure 6, Table 4, entries 1–3). The product **9a** with unexpected regiochemistry was formed with the best yield of 64% when only little excess of PhNH_2_ was applied. The reaction performs better in aprotic polar 1,4-dioxane (Table 4, entry 4) and worse in non-polar toluene or without a solvent (Table 4; entries 5, 6). It should be noted that an increase in the reaction temperature did not improve the yield of enaminone **9a**.

The reaction with electron-rich *p*-anisidine proceeded equally well, whereas electron-poor *p*-bromoaniline gave lower yield (Figure 7, Table 5). The reaction with aliphatic amines again has difficulties, probably due to the formation of salts; however, the lower acidity of TAL-*f*_3_ compared to DHA-*f*_6_ allowed enaminones **9d,e** to form in low yields. In the case of butylamine, the heating to 60 °C was also needed. All amines reacted with the same regioselectivity, and no alternative isomers **8** were isolated.

Although *Z*-form of enaminones is predominant because of an effective intramolecular hydrogen bonding, the contribution of an *E*-isomer may be affected by the electronic nature of a substituent at a nitrogen atom [36] or by an alternative hydrogen bond formation [37], which is the case for product **9** (Figure 3). The ratio of isomers is solvent-dependent, and *Z*-form mostly prevails, except for compound **4a** in DMSO*-d*_6_ (Table 5). The assignment of isomers was conducted on the basis of ^1^H and ^13^C NMR spectra (Appendix A). The key signals are low-field NH-protons of *Z*-**9** isomers at 12.39–12.58 ppm (for N-arylenaminones **9a**–**c** in CDCl_3_) and at 11.08–11.32 ppm (for N-alkylenaminones **9d**,**e** in CDCl_3_). The corresponding signals for the *E*-isomers of **9** are under a much higher field in accordance with the literature [37,38]. Among the other features of the NMR spectra of compounds **9** is the existence of methylene protons and amide NH-protons at a lower field and methyne carbons at a higher field for the *E*-isomer (Figure 3).

For an explanation of the difference in the regioselectivity of the interaction of TAL and TAL-*f*_3_ with amines, we propose the following mechanism. The addition of the first equivalent of RNH_2_ leads to the formation of dioxoamide **C** (Figure 8), which was isolated earlier (X = H) [39]. The intermediate **C** is then attacked by the second amine molecule at the less hindered atom, C-5, in the case of X = H forming product **8**. A preference of the attack at C-3 for the fluorinated derivative may be attributed to a high content of cyclic form **D** (Figure 8), analogs of which were described in the literature as the only tautomer in CDCl_3_ and DMSO-*d*_6_ solutions [40,41]. The semiaminal carbon atom is less susceptible to nucleophiles than the free keto group that leads to the selective formation of product **9**. It also should be noted that a mixture of regioisomers is usually produced when linear aliphatic CF_3_-diketones react with aniline [42].

No change in selectivity was observed when bifunctional aromatic amine, *o*-phenylenediamine, was used in the optimized conditions. Benzodiazepinone **10** was formed as the only isolated product, although the yield was moderate (Figure 9). Compound **10** was previously obtained from the corresponding dioxoester **11** in the mixture with diazepine **12** [43] (Figure 9), so our method represents a good alternative with no specific separation needed.

Phenylhydrazine also reacted with TAL-*f*_3_ (**2**) regioselectively in 1,4-dioxane, giving pyrazolohydrazide **13**, but the yield was poor. Changing the solvent to EtOH substantially increased the outcome, though isomer **14** was also formed and did not separate by the column chromatography (Figure 10). Compound **14** was previously synthesized from cyanopyrone **7** [35] and its spectral characteristics are in a good correlation with our data. Both isomers appear as two sets of signals corresponding to a major *syn*- (presented at Figure 10) and a minor *anti*-rotamer about the N–N bond. The key differences in the NMR spectra of the regioisomer **13** are the more downfield signal of the CF_3_ group in ^19^F NMR (106.2 ppm compared to 101.9 ppm for **14**) due to deshielding from the adjacent phenyl substituent, and more downfield signals of NH groups in ^1^H NMR (9.91 and 7.82 ppm compared to 9.84 and 7.73 ppm for **14**) due to additional hydrogen bonding.

Thus, hexafluorodehydroacetic acid and trifluorotriacetic acid lactone were shown to be active electrophiles and represent interesting fluorinated building blocks, which were transformed to a number of nitrogen heterocycles. Reaction of hexafluorodehydroacetic acid with primary aromatic amines leads to the formation of 2,6-bis(trifluoromethyl)-4-pyridones that are able to undergo Knoevenagel condensation to give merocyanine dyes. Trifluorotriacetic acid lactone undergoes ring-opening transformations with mono- and binucleophilic primary amines at the positions 2 and 4 and differs in regioselectivity compared to the non-fluorinated analog.

## 3. Materials and Methods

NMR spectra were recorded on Bruker DRX-400 (Bruker BioSpin GmbH, Ettlingen, Germany, work frequencies: ^1^H–400 MHz, ^13^C–100 MHz, ^19^F–376.5 MHz) and Bruker Avance III-500 (Bruker BioSpin GmbH, Rheinstetten, Germany, work frequencies: ^1^H–500 MHz, ^13^C–126 MHz, ^19^F–471 MHz) spectrometers in DMSO*-d*_6_ and CDCl_3_. The chemical shifts (*δ*) are reported in ppm relative to the internal standard TMS (^1^H NMR), C_6_F_6_ (^19^F NMR), and residual signals of the solvents (^13^C NMR). IR spectra were recorded on a Shimadzu IRSpirit-T spectrometer (Shimadzu Corp., Kyoto, Japan) using an attenuated total reflectance (ATR) unit (FTIR mode, ZnSe crystal), and the absorbance maxima (*ν*) are reported in cm^−1^. UV-visible spectra were recorded on a Shimadzu UV-1900 spectrophotometer (Shimadzu Corp., Kyoto, Japan) using EtOH as a solvent, the absorbance maxima (*λ*) are reported in nm, and the molar attenuation coefficients are reported in L·mol^−1^·cm^−1^. Elemental analyses were performed on an automatic analyzer PerkinElmer PE 2400 Series II (Perkin Elmer Instruments, Waltham, MA, USA). Melting points were determined using a Stuart SMP40 melting point apparatus (Bibby Scientific Ltd., Stone, Staffordshire, UK). Column chromatography was performed on silica gel (Merck 60, 70–230 mesh). All solvents and reagents were obtained commercially and used without purification.

### 3.1. Synthesis of Pyrones ***1*** and ***2***

*4-Hydroxy-3-(2,2,2-trifluoroacetyl)-6-(trifluoromethyl)-2H-pyran-2-one* (**1**). A mixture of ethyl 4,4,4-trifluoroacetoacetate (9.1 g, 0.032 mol) and P_2_O_5_ (40 g, 0.141 mol) was heated with a gas burner flame for 10 min. Then, the mixture was distilled in a vacuum (~20 torr), collecting the fraction 90–110 °C. The target fraction was redistilled in the same interval. Yield 4.1 g (75%), yellowish needles, mp 40–42 °C. ^1^H NMR (500 MHz, DMSO*-d*_6_) *δ* 6.24 (1H, s, H-5), 11.12 (1H, s, OH). ^19^F NMR (471 MHz, DMSO*-d*_6_) *δ* 89.6 (s, COCF_3_), 91.1 (s, 6-CF_3_) [25].

*4-Hydroxy-6-(trifluoromethyl)-2H-pyran-2-one* (**2**). A saturated aqueous solution of NaHCO_3_ (0.5 mL, 0.55 mmol) was added with stirring to an aqueous solution of hexafluorodehydroacetic acid (**1**) (2.5 mL, 1 M). The mixture was stirred for 10 min and acidified with aqueous HCl (1 mL, 3 M). The precipitate was filtered off and dried. Yield 180 mg (40%), white solid, mp 135–138 °C (lit. mp 135–137 °C [25]). ^1^H NMR (500 MHz, DMSO*-d*_6_) *δ* 5.59 (1H, d, *J* = 1.8 Hz, H-3), 6.83 (1H, d, *J* = 1.7 Hz, H-5), 12.60 (1H, s, OH). ^19^F NMR (471 MHz, DMSO*-d*_6_) *δ* 92.0 (s, CF_3_).

### 3.2. Synthesis of Compounds ***3a***–***k***

#### General Procedure

An aromatic amine (0.83 mmol) was added to a solution of hexafluorodehydroacetic acid (**1**) (100 mg, 0.36 mmol) in EtOH (1 mL). The reaction mixture was stirred at room temperature for a given amount of time and acidified with aqueous HCl (3 mL, 1 M). The precipitate was filtered off and washed with water. The crude product was recrystallized from hexane.

*1-Phenyl-2,6-bis(trifluoromethyl)pyridin-4(1H)-one* (**3a**). The reaction was carried out for 24 h. Yield 66 mg (60%), yellow powder, mp 135–136 °C. ^1^H NMR (400 MHz, CDCl_3_) *δ* 6.97 (2H, s, H-3, H-5), 7.39 (2H, d, *J* = 7.6 Hz, H Ph), 7.50 (2H, t, *J* = 7.9 Hz, H Ph), 7.59 (1H, t, *J* = 7.5 Hz, H Ph). ^1^H NMR (500 MHz, DMSO*-d*_6_) *δ* 6.97 (2H, s, H-3, H-5), 7.55 (2H, t, *J* = 7.7 Hz, H Ph), 7.59 (1H, t, *J* = 7.5 Hz, H Ph), 7.70 (2H, d, *J* = 7.8 Hz, H Ph). ^19^F NMR (471 MHz, CDCl_3_) *δ* 100.6 (s, CF_3_). ^19^F NMR (471 MHz, DMSO*-d*_6_) *δ* 102.5 (s, CF_3_). ^13^C NMR (126 MHz, CDCl_3_) *δ* 119.1 (2C, q, *J* = 4.7 Hz, C-3, C-5), 119.4 (2C, q, *J* = 275.7 Hz, CF_3_), 128.7 (2C Ph), 130.1 (2C Ph), 131.3 (C Ph), 135.0 (C Ph), 141.2 (2C, q, *J* = 33.3 Hz, C-2, C-6), 177.4 (CO). IR (ATR) ν 3297, 1700 (C=O), 1658, 1398 (N–C), 1128–1116 (CF_3_). Anal. Calculated for C_13_H_7_F_6_NO: C 50.83; H 2.30; N 4.56. Found: C 50.77; H 2.38; N 4.46.

*1-(4-Methoxyphenyl)-2,6-bis(trifluoromethyl)pyridin-4(1H)-one* (**3b**). The reaction was carried out for 8 h. Yield 68 mg (56%), gray powder, mp 125–126 °C. ^1^H NMR (400 MHz, CDCl_3_) *δ* 3.88 (3H, s, CH_3_), 6.95 (2H, s, H-3, H-5), 6.95 (2H, d, *J* = 8.9 Hz, H Ar), 7.28 (2H, d, *J* = 8.9 Hz, H Ar). ^19^F NMR (471 MHz, CDCl_3_) *δ* 100.5 (s, CF_3_). ^13^C NMR (126 MHz, CDCl_3_) *δ* 55.6 (OCH_3_), 113.7 (2C Ar), 119.1 (2C, q, *J* = 4.7 Hz, C-3, C-5), 119.4 (2C, q, *J* = 275.6 Hz, CF_3_), 127.2 (C Ar), 131.2 (2C Ar), 141.6 (2C, q, *J* = 32.7 Hz, C-2, C-6), 161.3 (C Ar), 177.5 (CO). IR (ATR) ν 3061, 1653 (C=O), 1602, 1510, 1397 (N–C), 1250, 1397, 1158 (CF_3_). HRMS (ESI) m/z [M + H]^+^. Calculated for C_14_H_10_F_6_NO_2_: 338.0616. Found: 338.0613.

*1-(3,4-Difluorophenyl)-2,6-bis(trifluoromethyl)pyridin-4(1H)-one* (**3c**). The reaction was carried out for 6 h. Yield 71 mg (57%), gray powder, mp 142–143 °C. ^1^H NMR (500 MHz, CDCl_3_) *δ* 6.96 (2H, s, H-3, H-5), 7.21 (1H, d, *J* = 8.6 Hz, H Ar), 7.31 (2H, m, H Ar). ^19^F NMR (471 MHz, CDCl_3_) *δ* 28.3 (1F, dt, *J* = 21.3, 8.8 Hz, F Ar), 31.0 (1F, dddd, *J* = 21.3, 9.5, 6.7, 3.8 Hz, F Ar), 100.7 (6F, s, CF_3_). ^13^C NMR (101 MHz, CDCl_3_) *δ* 117.4 (d, *J* = 18.9 Hz, C Ar), 119.3 (2C, q, *J* = 275.7 Hz, CF_3_), 119.4 (2C, q, *J* = 4.7 Hz, C-3, C-5), 120.2 (d, *J* = 19.1 Hz, C Ar), 127.3 (C Ar), 130.6 (C Ar), 140.9 (2C, q, *J* = 33.3 Hz, C-2, C-6), 149.5 (dd, *J* = 254.1, 13.8 Hz, C Ar), 152.1 (dd, *J* = 256.4, 12.2 Hz, C Ar), 177.1 (CO). IR (ATR) ν 3091, 3056, 1655 (C=O), 1615, 1519, 1397 (N–C), 1144 (CF_3_). Anal. Calculated for C_13_H_5_F_8_NO: C 45.50; H 1.47; N 4.08. Found: C 45.33; H 1.46; N 4.09.

*1-(4-Bromophenyl)-2,6-bis(trifluoromethyl)pyridin-4(1H)-one* (**3d**). The reaction was carried out for 24 h. Yield 65 mg (47%), yellow powder, mp 129–131 °C (decomp.). ^1^H NMR (400 MHz, CDCl_3_) *δ* 6.97 (2H, s, H-3, H-5), 7.27 (2H, d, *J* = 8.1 Hz, H Ar), 7.64 (1H, d, *J* = 8.5 Hz, H Ar). ^19^F NMR (471 MHz, CDCl_3_) *δ* 100.7 (s, CF_3_). ^13^C NMR (101 MHz, CDCl_3_) *δ* 119.26 (2C, q, *J* = 4.5 Hz, C-3, C-5), 119.33 (2C, q, *J* = 275.8 Hz, CF_3_), 126.0 (C Ar), 131.7 (2C, C Ar), 132.1 (2C, C Ar), 133.9 (C Ar), 141.0 (2C, q, *J* = 32.8 Hz, C-2, C-6), 177.3 (CO). IR (ATR) ν 3071, 1652 (C=O), 1589, 1480, 1397 (N–C), 1249, 1175 (CF_3_). HRMS (ESI) m/z [M + H]^+^. Calculated for C_13_H_7_BrF_6_NO: 385.9615. Found: 385.9606.

*1-(3-Acetylphenyl)-2,6-bis(trifluoromethyl)pyridin-4(1H)-one* (**3e**). The reaction was carried out for 24 h. Yield 60 mg (47%), pale pink powder, mp 118–119 °C. ^1^H NMR (400 MHz, CDCl_3_) *δ* 2.65 (3H, s, CH_3_), 6.99 (2H, s, H-3, H-5), 7.57–7.69 (2H, m, H Ar), 8.00 (1H, s, H Ar), 8.17 (1H, dt, *J* = 7.4, 1.5 Hz, H Ar). ^19^F NMR (376 MHz, CDCl_3_) *δ* 100.8 (s, CF_3_). ^13^C NMR (126 MHz, CDCl_3_) *δ* 26.6 (CH_3_), 119.0 (2C, unres. q, C-3, C-5), 119.2 (2C, q, *J* = 276.1 Hz, CF_3_), 129.2 (C Ar), 129.8 (C Ar), 131.2 (C Ar), 134.1 (C Ar), 135.4 (C Ar), 137.7 (C Ar), 141.6 (2C, unres. q, C-2, C-6), 177.2 (4-CO), 195.6 (COCH_3_). IR (ATR) ν 3068, 1700 (C=O), 1665 (C=O), 1610, 1480, 1402 (N–C), 1138 (CF_3_). Anal. Calculated for C_15_H_9_F_6_NO_2_: C 51.59; H 2.60; N 4.01. Found: C 51,50; H 2.63; N 3.99.

*1-(3-(Trifluoromethyl)phenyl)-2,6-bis(trifluoromethyl)pyridin-4(1H)-one* (**3f**). The reaction was carried out for 24 h. Yield 63 mg (47%), light-yellow powder, mp 91–92 °C. ^1^H NMR (600 MHz, CDCl_3_) *δ* 7.00 (2H, s, H-3, H-5), 7.64 (1H, d, *J* = 8.2 Hz, H Ar), 7.69 (1H, t, *J* = 8.0 Hz, H Ar), 7.72 (1H, s, H Ar), 7.89 (1H, d, *J* = 7.8 Hz, H Ar). ^19^F NMR (376 MHz, CDCl_3_) *δ* 98.7 (3F, s, CF_3_ Ar), 100.7 (6F, s, 2-CF_3_, 6-CF_3_). ^13^C NMR (126 MHz, CDCl_3_) *δ* 119.3 (2C, q, *J* = 275.5 Hz, 2-CF_3_, 6-CF_3_), 119.4 (2C, q, *J* = 4.7 Hz, C-3, C-5), 122.9 (q, *J* = 272.5 Hz, CF_3_ Ar), 127.6 (C Ar), 128.3 (C Ar), 129.6 (C Ar), 131.8 (q, *J* = 33.8 Hz, C-3 Ar), 133.6 (C Ar), 135.4 (C Ar), 140.9 (2C, q, *J* = 33.4 Hz, C-2, C-6), 177.2 (CO). IR (ATR) ν 3091, 1655 (C=O), 1395 (N–C), 1327, 1252, 1126 (CF_3_). HRMS (ESI) m/z [M + H]^+^. Calculated for C_14_H_7_F_9_NO: 376.0384. Found: 376.0397.

*1-(2,5-Difluorophenyl)-2,6-bis(trifluoromethyl)pyridin-4(1H)-one* (**3g**). The reaction was carried out for 48 h. Yield 58 mg (46%), yellow powder, mp 116–119 °C. ^1^H NMR (500 MHz, CDCl_3_) *δ* 6.99 (2H, s, H-3, H-5), 7.20–7.26 (2H, m, H-3, H-6 Ar), 7.35 (1H, dddd, *J* = 9.3, 7.4, 3.8, 3.0 Hz, H-4 Ar). ^19^F NMR (376 MHz, CDCl_3_) *δ* 38.6–38.8 (m, F-2 Ar), 46.0 (1F, dtd, *J* = 15.4, 7.4, 4.7 Hz, F-5 Ar), 98.5 (dd, *J* = 3.2, 0.9 Hz, CF_3_). ^13^C NMR (126 MHz, CDCl_3_) *δ* 117.1 (dd, *J* = 22.4, 8.8 Hz, C Ar), 119.1 (d, *J* = 25.9 Hz, C Ar), 119.3 (2C, q, *J* = 275.5 Hz, CF_3_), 119.5 (2C, q, *J* = 4.6 Hz, C-3, C-5), 120.8 (dd, *J* = 23.6, 8.1 Hz, C Ar), 123.4 (dd, *J* = 16.3, 10.5 Hz, C Ar), 140.8 (2C, q, *J* = 34.1 Hz, C-2, C-6), 155.9 (dd, *J* = 252.0, 3.3 Hz, C Ar), 157.4 (dd, *J* = 247.8, 2.8 Hz, C Ar), 177.2 (CO). IR (ATR) ν 3072, 1666 (C=O), 1615, 1508, 1397 (N–C), 1251, 1136 (CF_3_). HRMS (ESI) m/z [M + H]^+^. Calculated for C_13_H_6_F_8_NO: 344.0322. Found: 344.0316.

*1-(2-Chlorophenyl)-2,6-bis(trifluoromethyl)pyridin-4(1H)-one* (**3h**). The reaction was carried out for 40 h. Yield is 61 mg (49%), yellow powder, mp 132–136 °C. ^1^H NMR (500 MHz, CDCl_3_) *δ* 6.99 (2H, s, H-3, H-5), 7.41–7.48 (1H, m, H Ar), 7.51–7.58 (3H, m, H Ar). ^19^F NMR (471 MHz, CDCl_3_) *δ* 98.6 (s, CF_3_). ^13^C NMR (126 MHz, CDCl3) *δ* 119.3 (2C, q, *J* = 275.8 Hz, CF_3_), 119.6 (2C, q, *J* = 4.5 Hz, C-3, C-5), 127.2 (C Ar), 130.1 (C Ar), 131.9 (C Ar), 132.7 (C Ar), 132.9 (C Ar), 135.9 (C Ar), 140.9 (2C, q, *J* = 33.9 Hz, C-2, C-6), 177.6 (CO). IR (ATR) ν 3056, 1743, 1651 (C=O), 1476, 1389 (N–C), 1247, 1136 (CF_3_). HRMS (ESI) m/z [M + H]^+^. Calculated for C_13_H_7_ClF_6_NO: 342.0120. Found: 342.0131.

*1-(o-Tolyl)-2,6-bis(trifluoromethyl)pyridin-4(1H)-one* (**3i**). The reaction was carried out for 24 h. Yield 74 mg (65%), brown viscous liquid. ^1^H NMR (400 MHz, CDCl_3_) *δ* 2.08 (3H, s, CH_3_), 7.01 (2H, s, H-3, H-5), 7.35–7.30 (3H, m, H Ar), 7.47 (1H, td, *J* = 7.5, 1.6 Hz, H Ar). ^19^F NMR (471 MHz, CDCl_3_) *δ* 99.0 (d, *J* = 1.0 Hz, CF_3_). ^13^C NMR (151 MHz, CDCl_3_) *δ* 17.0 (CH_3_), 119.3 (2C, q, *J* = 275.8 Hz, CF_3_), 119.6 (2C, q, *J* = 4.2 Hz, C-3, C-5), 126.2 (C Ar), 130.3 (C Ar), 130.7 (C Ar), 131.5 (C Ar), 134.1 (C Ar), 138.5 (C Ar), 141.3 (2C, q, *J* = 33.2 Hz, C-2, C-6), 177.9 (CO). IR (ATR) ν 3054, 1696 (C=O), 1620, 1508, 1391 (N–C), 1252, 1135 (CF_3_). HRMS (ESI) m/z [M + H]^+^. Calculated for C_14_H_10_F_6_NO: 322.0667. Found: 322.0679.

*1-(2,5-Dimethylphenyl)-2,6-bis(trifluoromethyl)pyridin-4(1H)-one* (**3j**). The reaction was carried out for 24 h. Yield 58 mg (48%), brown viscous liquid, crystallizes on standing, mp 105–107 °C. ^1^H NMR (500 MHz, CDCl_3_) *δ* 2.02 (3H, s, CH_3_) 2.38 (3H, s, CH_3_), 7.01 (2H, s, H-3, H-5), 7.16 (1H, s, H-2 Ar), 7.18 (1H, d, *J* = 7.7 Hz, H-5 Ar), 7.27 (1H, d, *J* = 7.7 Hz, H-4 Ar). ^19^F NMR (471 MHz, CDCl_3_) *δ* 99.0 (s, CF_3_). ^13^C NMR (126 MHz, CDCl_3_) *δ* 16.6 (CH_3_), 20.6 (CH_3_), 119.3 (2C, q, *J* = 275.8 Hz, CF_3_), 119.5 (2C, q, *J* = 4.8 Hz, C-3, C-5), 130.3 (C Ar), 130.6 (C Ar), 132.2 (C Ar), 133.9 (C Ar), 135.2 (C Ar), 136.2 (C Ar), 141.1 (2C, q, *J* = 33.4 Hz, C-2, C-6), 177.5 (CO). IR (ATR) ν 3091, 1655 (C=O), 1608, 1395 (N–C), 1328, 1251, 1125 (CF_3_). HRMS (ESI) m/z [M + H]^+^. Calculated for C_15_H_12_F_6_NO: 336.0823. Found: 336.0802.

*1-(4-Nitrophenyl)-2,6-bis(trifluoromethyl)pyridin-4(1H)-one* (**3k**). The reaction was carried out for 6 d. Pale yellow powder, a mixture of **3k** (10 mg, 8%) and 4-nitroaniline (32 mg). ^1^H NMR (500 MHz, CDCl_3_) *δ* 7.00 (2H, s, H-3, H-5), 7.64 (2H, d, *J* = 8.4 Hz, H Ar), 8.39 (2H, d, *J* = 8.9 Hz, H Ar). ^19^F NMR (471 MHz, CDCl_3_) *δ* 100.9 (s, CF_3_). HRMS (ESI) m/z [M + H]^+^. Calculated for C_13_H_7_F_6_N_2_O_3_: 353.0361. Found: 353.0357.

### 3.3. Synthesis of Compound ***B′***

*o*-Chloroaniline (106 mg, 0.83 mmol) was added to a solution of hexafluorodehydroacetic acid (**1**) (100 mg, 0.36 mmol) in EtOH (1 mL). The reaction mixture was stirred at room temperature for a 15 h and acidified with aqueous HCl (3 mL, 1 M). The precipitate was filtered off and washed with water. The crude product was purified by column chromatography (CHCl_3_).

*(2Z,5Z)-2,6-Bis((2-chlorophenyl)amino)-1,1,1,7,7,7-hexafluorohepta-2,5-dien-4-one* (**B′**). Dark-yellow solid, a mixture of **B′** (108 mg, 64%) and 2-chloroaniline (11 mg), mp 84–85 °C. ^1^H NMR (400 MHz, CDCl_3_) *δ* 5.78 (2H, s, 3-CH, 5-CH), 7.18–7.29 (4H, m, H Ar), 7.33 (2H, d, *J* = 7.3 Hz, H Ar), 7.44 (2H, dd, *J* = 7.8, 1.5 Hz, H Ar), 11.49 (2H, s, NH). ^19^F NMR (376 MHz, CDCl_3_) *δ* 97.8 (s, CF_3_).

### 3.4. Synthesis of Compound ***4***

2-Aminopyridine (78 mg, 0.83 mmol) was added to a solution of hexafluorodehydroacetic acid (**1**) (100 mg, 0.36 mmol) in EtOH (1 mL). The precipitate was filtered and washed with a small amount of EtOH.

*2-Aminopyridin-1-ium 2-oxo-3-(2,2,2-trifluoroacetyl)-6-(trifluoromethyl)-2H-pyran-4-olate* (**4**). Yield 127 mg (95%), white solid, mp 205–206 °C. ^1^H NMR (400 MHz, DMSO-d_6_) δ 6.15 (1H, s, H-5 pyrone), 6.86 (1H, td, J = 6.8, 1.0 Hz, H-5 pyridine), 6.97 (1H, dd, J = 9.6, 1.0 Hz, H-3 pyridine), 7.92 (4H, m, NH_2_, H-4, H-6 pyridine), 13.27 (1H, s, NH). ^19^F NMR (376 MHz, DMSO-d_6_) δ 89.8 (s, COCF_3_), 91.1 (s, CF_3_). ^13^C NMR (126 MHz, DMSO-d_6_) δ 98.4 (C-3), 109.7 (q, J = 2.9 Hz, C-5), 112.1 (C Py), 113.4 (C Py), 116.6 (q, J = 292.1 Hz, CF_3_), 118.9 (q, J = 272.8 Hz, CF_3_), 136.0 (C Py), 144.1 (C Py), 146.8 (q, J = 37.1 Hz, C-6), 153.9 (C Py), 160.0 (C-2), 176.8 (C-4), 177.7 (q, J = 33.5 Hz, COCF_3_). IR (ATR) ν 3366, 3310, 3191, 1712 (C=O), 1699 (C=O), 1535, 1372 (N–C), 1134 (CF_3_). Anal. Calculated for C_13_H_8_F_6_N_2_O_4_: C 42.18; H 2.18; N 7.57. Found: C 42.28; H 2.07; N 7.57.

### 3.5. Synthesis of Compounds ***5***

#### General Procedure

A mixture of 1-phenyl-2,6-bis(trifluoromethyl)pyridin-4(1*H*)-one (100 mg, 0.33 mmol) and a corresponding active methylene compound (0.4 mmol) in Ac_2_O (1.5 mL) was heated at 140 °C for a given amount of time. The reaction mixture was then cooled and left overnight for crystallization of a product. The precipitate was filtered and washed with EtOH (1 mL). The product was dried at 120 °C for 4 h.

*2-(1-Phenyl-2,6-bis(trifluoromethyl)pyridin-4(1H)-ylidene)malononitrile* (**5a**). Synthesized from malononitrile (10 h). Yield 77 mg (65%). Dark-yellow crystals, mp 172–173 °C. ^1^H NMR (500 MHz, DMSO-*d_6_*) *δ* 7.26 (2H, s, H-3′, H-5′), 7.59 (2H, t, *J* = 7.8 Hz, H Ph), 7.66 (1H, t, *J* = 7.6 Hz, H Ph), 7.74 (2H, d, *J* = 7.8 Hz, H Ph). ^19^F NMR (471 MHz, DMSO-*d_6_*) *δ* 102.5 (s, CF_3_). ^13^C NMR (126 MHz, DMSO-*d_6_*) *δ* 57.1 (C(CN)_2_), 112.7 (2C, q, *J* = 5.3 Hz, C-3′, C-5′), 114.8 (2C, CN), 118.7 (2C, q, *J* = 275.9 Hz, CF_3_), 128.7 (2C Ph), 130.1 (2C Ph), 131.7 (C Ph), 134.6 (C Ph), 138.2 (2C, q, *J* = 33.0 Hz, C-2′, C-6′), 154.2 (C-4′). IR (ATR) ν 3058, 2922, 2852, 2215, 1643, 1518, 1302 (N–C), 1275, 1142 (CF_3_). UV/vis (EtOH): *λ*_max_ (ε_max_) = 382 (18222). HRMS (ESI) m/z [M + H]^+^. Calculated for C_16_H_8_F_6_N_3_: 356.0622. Found: 356.0624.

5-(1-Phenyl-2,6-bis(trifluoromethyl)pyridin-4(1H)-ylidene)pyrimidine-2,4,6(1H,3H,5H)-trione (**5b**). Synthesized from barbituric acid (4 h). Yield 81 mg (58%). Dark-yellow powder, mp 270–271 °C (decomp.). ^1^H NMR (500 MHz, DMSO-*d_6_*) *δ* 7.59 (2H, t, *J* = 7.7 Hz, H Ph), 7.66 (1H, t, *J* = 7.5 Hz, H Ph), 7.80 (2H, d, *J* = 7.9 Hz, H Ph), 9.77 (2H, s, H-3′, H-5′), 10.78 (2H, s, NH). ^19^F NMR (471 MHz, DMSO-*d_6_*) *δ* 103.2 (s, CF_3_). ^13^C NMR (151 MHz, DMSO-*d_6_*) *δ* 95.0 (C-5), 116.8 (2C, q, *J* = 5.7 Hz, C-3′, C-5′), 119.8 (2C, q, *J* = 275.5 Hz, CF_3_), 129.0 (2C Ph), 130.4 (2C Ph), 132.1 (C Ph), 135.6 (C Ph), 138.3 (2C, q, *J* = 32.3 Hz, C-2′, C-5′), 150.1 (2-CO), 153.0 (C-4′), 165.6 (2C, 4-CO, 6-CO). IR (ATR) ν 3021, 1712, 1673, 1483, 1354 (N–C), 1145 (CF_3_). UV/vis (EtOH): λ_max_ (ε_max_) = 412 (91406), 234 (37760). HRMS (ESI) m/z [M + H]^+^. Calculated for C_17_H_10_F_6_N_3_O_3_: 418.0626. Found: 418.0591.

2-(1-Phenyl-2,6-bis(trifluoromethyl)pyridin-4(1H)-ylidene)1,3-indandione (**5c**). Synthesized from 1,3-indandione (6 h). Yield 62 mg (43%). Dark-yellow crystals, mp 208–209 °C. ^1^H NMR (500 MHz, DMSO-*d_6_*) *δ* 7.60 (2H, t, *J* = 7.7 Hz, H Ph), 7.67 (1H, t, *J* = 7.4 Hz, H Ph), 7.70–7.78 (4H, m, H Ar), 7.80 (2H, d, *J* = 7.9 Hz, H Ph), 9.10 (2H, s, H-3′, H-5′). ^19^F NMR (471 MHz, DMSO-*d_6_*) *δ* 102.8 (s, CF_3_). ^13^C NMR (126 MHz, DMSO-*d_6_*) *δ* 107.2 (C-2), 113.4 (2C, q, *J* = 5.9 Hz, C-3′, C-5′), 119.2 (2C, q, *J* = 275.2 Hz, CF_3_), 121.3 (2C Ar), 128.5 (2C Ph), 130.0 (2C Ph), 131.6 (C Ph), 134.0 (2C Ar), 135.1 (C Ph), 138.9 (2C, q, *J* = 32.7 Hz, C-2′, C-6′), 139.8 (2C Ar), 146.0 (C-4′), 191.0 (2C, CO). IR (ATR) ν 3101, 3065, 1693, 1648, 1518, 1319 (N–C), 1141 (CF_3_). UV/vis (EtOH): λ_max_ (ε_max_) = 437 (21948), 415 (15988), 246 (74128), 224 (151744). HRMS (ESI) m/z [M + H]^+^. Calculated for C_22_H_12_F_6_NO_2_: 436.0772. Found: 436.0779.

### 3.6. Synthesis of Compounds ***9***

#### General Procedure

To a solution of trifluorotriacetic acid (**2**) (100 mg, 0.56 mmol) in 1,4-dioxane (1 mL), a corresponding amine (1.17 mmol) was added with stirring. The reaction was monitored by TLC (CHCl_3_:EtOH, 20:1). After completion, the reaction mixture was acidified with aqueous HCl (1 mL, 3 M). The precipitate was filtered and washed with water. The crude product was recrystallized from hexane.

*6,6,6-Trifluoro-5-oxo-N-phenyl-3-(phenylamino)hex-3-enamide* (**9a**). The reaction was carried out for 24 h. Yield 139 mg (72%), small yellow crystals, mp 135–137 °C. ^1^H NMR (500 MHz, DMSO*-d*_6_) *δ Z-***9a** (44%): 3.65 (2H, s, 2-CH_2_), 5.79 (1H, s, 4-CH), 7.05 (2H, t, *J* = 7.3 Hz, H Ph), 7.30–7.34 (2H, m, H Ph), 7.40 (2H, d, *J* = 7.5 Hz, H Ph), 7.45 (4H, t, *J* = 7.0 Hz, H Ph), 10.08 (1H, s, NH amide), 12.56 (1H, s, NH enamine); *E-***9a** (56%): 4.06 (2H, s, 2-CH_2_), 5.59 (1H, s, 4-CH), 7.25–7.38 (6H, m, H Ph), 7.52 (2H, t, *J* = 6.8 Hz, H Ph), 7.61 (2H, d, *J* = 7.4 Hz, H Ph), 10.23 (1H, s, NH), 10.31 (1H, s, NH). ^19^F NMR (376 MHz, DMSO*-d*_6_) *δ Z-***9a** (44%): 87.1 (s, CF_3_); *E-***9a** (56%): 86.6 (s, CF_3_). ^19^F NMR (376 MHz, CDCl_3_) *δ Z-***9a** (59%): 85.1 (s, CF_3_); *E-***9a** (41%): 84.8 (s, CF_3_). ^13^C NMR (101 MHz, CDCl_3_) *δ Z-***9a** (59%): 41.5 (CH_2_), 91.3 (4-CH), 117.2 (q, *J* = 288.3 Hz, CF_3_), 120.2 (2C Ph), 125.2 (C Ph), 125.9 (2C Ph), 128.3 (C Ph), 129.1 (2C Ph), 129.82 (2C Ph), 136.4 (C Ph), 136.9 (C Ph) 163.4 (1-CO), 164.0 (C-3), 177.8 (q, *J* = 33.7 Hz, 5-CO); *E-***9a** (41%): 42.3 (CH_2_), 88.7 (CH), 117.6 (q, *J* = 290.2 Hz, CF_3_), 120.4 (2C Ph), 124.7 (2C Ph), 124.9 (C Ph), 127.8 (C Ph), 129.0 (2C Ph), 129.78 (2C Ph), 136.6 (C Ph), 137.5 (C Ph), 162.5 (1-CO), 166.3 (C-3), 178.5 (q, *J* = 32.1 Hz, 5-CO). IR (ATR) ν 3299, 3061, 3032, 1658 (C=O), 1578, 1242, 1172, 1116 (CF_3_). Anal. Calculated for C_18_H_15_F_3_N_2_O_2_: C 62.07; H 4.34; N 8.04. Found: C 61.81; H 4.12; N 7.94.

*6,6,6-Trifluoro-N-(4-methoxyphenyl)-3-((4-methoxyphenyl)amino)-5-oxohex-3-enamide* (**9b**). The reaction was carried out for 17 h. Yield 160 mg (69%), fine gray crystals, mp 163–165 °C. ^1^H NMR (400 MHz, CDCl_3_) *δ Z-***9b** (65%): 3.38 (2H, s, 2-CH_2_), 3.79 (3H, s, OCH_3_) 3.80 (3H, s, OCH_3_), 5.68 (1H, s, 4-CH), 6.85 (2H, d, *J* = 8.7 Hz, H Ar), 6.91 (2H, d, *J* = 8.6 Hz, H Ar), 7.16 (1H, d, *J* = 8.6 Hz, H Ar), 7.29 (1H, d, *J* = 8.7 Hz, H Ar), 7.12 (1H, s, NH amide), 12.39 (1H, s, NH enamine); *E-***9b** (35%): 3.79 (3H, s, OCH_3_) 3.83 (3H, s, OCH_3_), 4.00 (2H, s, 2-CH_2_), 5.68 (1H, s, 4-CH), 6.84 (2H, d, *J* = 8.7 Hz, H Ar), 6.90 (2H, d, *J* = 8.6 Hz, H Ar), 7.11 (1H, d, *J* = 8.6 Hz, H Ar), 7.42 (1H, d, *J* = 8.7 Hz, H Ar), 8.74 (1H, s, NH amide), 9.63 (1H, s, NH enamine). ^19^F NMR (376 MHz, CDCl_3_) *δ Z-***9b** (65%): 85.1 (s, CF_3_); *E-***9b** (35%): 84.9 (s, CF_3_). ^13^C NMR (126 MHz, CDCl_3_) *δ Z-***9b** (65%): 41.4 (CH_2_), 55.47 (OCH_3_), 55.52 (OCH_3_), 90.9 (4-CH), 114.3 (2C Ar), 114.88 (2C Ar), 117.3 (q, *J* = 288.6 Hz, CF_3_), 122.1 (2C Ar), 127.4 (2C Ar), 129.0 (C Ar), 129.9 (C Ar), 157.1 (C Ar), 159.4 (C Ar), 164.0, 164.2, 177.5 (q, *J* = 33.6 Hz, 5-CO); *E-***9b** (35%): 42.0 (CH_2_), 55.47 (OCH_3_), 55.49 (OCH_3_), 88.2 (4-CH), 114.1 (2C Ar), 114.91 (2C Ar), 117.7 (q, *J* = 290.2 Hz, CF_3_), 122.0 (2C Ar), 126.4 (2C Ar), 129.3 (C Ar), 130.8 (C Ar), 156.7 (C Ar), 159.0 (C Ar), 163.2 (1-CO), 165.9 (C-3), 177.9 (q, *J* = 31.3 Hz, 5-CO). IR (ATR) ν 3282, 2999, 2842, 1651 (C=O), 1530, 1237, 1169, 1116 (CF_3_). Anal. Calculated for C_20_H_19_F_3_N_2_O_4_**·**0.25H_2_O: C 58.18; H 4.76; N 6.79. Found: C 58.25; H 4.48; N 6.54.

*N-(4-Bromophenyl)-3-((4-bromophenyl)amino)-6,6,6-trifluoro-5-oxohex-3-enamide* (**9c**). The reaction was carried out for 15 h. Yield 118 mg (41%), fine yellow crystals, mp 209–210 °C (decomp.). ^1^H NMR (500 MHz, CDCl_3_) *δ Z-***9c** (58%): 3.41 (2H, s, 2-CH_2_), 5.70 (1H, s, 4-CH), 7.14 (2H, d, *J* = 8.4 Hz, H Ar), 7.31 (2H, d, *J* = 8.6 Hz, H Ar), 7.43 (1H, s, NH amide), 7.46 (2H, d, *J* = 8.6 Hz, H Ar), 7.54 (2H, d, *J* = 8.4 Hz, H Ar), 12.42 (1H, s, NH enamine); *E-***9c** (42%): 3.98 (1H, s, 2-CH_2_), 5.78 (1H, s, 4-CH), 7.09 (2H, d, *J* = 8.5 Hz, H Ar), 7.41–7.46 (4H, m, H Ar), 7.55 (2H, d, *J* = 8.5 Hz, H Ar), 8.32 (1H, s, NH amide), 9.62 (1H, s, NH enamine). ^19^F NMR (471 MHz, CDCl_3_) *δ Z-***9c** (58%): 85.0 (s, CF_3_); *E-***9c** (42%): 84.8 (s, CF_3_). IR (ATR) ν 3262, 3018, 3046, 1664 (C=O), 1589, 1533, 1246, 1111 (CF_3_). Anal. Calculated for C_18_H_13_Br_2_F_3_N_2_O_2_**·**0.33H_2_O: C 42.22; H 2.69; N 5.47. Found: C 42.44; H 2.58; N 5.19.

*N-Benzyl-3-(benzylamino)-6,6,6-trifluoro-5-oxohex-3-enamide* (**9d**). The reaction was carried out for 17 h. Yield 51 mg (24%), colorless powder, mp 124–125 °C. ^1^H NMR (500 MHz, DMSO*-d*_6_) *δ Z-***9d** (68%**)**: 3.55 (2H, s, 2-CH_2_), 4.30 (2H, d, *J* = 5.8 Hz, NCH_2_), 4.72 (2H, d, *J* = 6.1 Hz, NCH_2_), 5.50 (1H, s, 4-CH), 7.09–7.49 (10H, m, H Ph), 8.76 (1H, t, *J* = 5.8 Hz, NH amide), 11.29 (1H, t, *J* = 6.1 Hz, NH enamine); *E-***9d** (32%): 3.84 (2H, s, 2-CH_2_), 4.31 (2H, d, *J* = 6.0 Hz, NCH_2_), 4.46 (2H, d, *J* = 5.6 Hz, NCH_2_), 5.27 (1H, s, 4-CH), 7.09–7.49 (10H, m, H Ph), 8.45 (1H, t, *J* = 5.9 Hz, NH amide), 9.08 (1H, t, *J* = 5.8 Hz, NH enamine). ^1^H NMR (500 MHz, CDCl_3_) *δ Z-***9d** (58%): 3.31 (2H, s, 2-CH_2_), 4.39 (2H, d, *J* = 5.7 Hz, NCH_2_), 4.61 (2H, d, *J* = 6.1 Hz, NCH_2_), 5.44 (1H, s, 4-CH), 6.04 (1H, unres. t, NH) 7.20–7.40 (10H, m, H Ph), 11.32 (1H, unres. t, NH); *E-***9d** (42%): 3.79 (2H, s, 2-CH_2_), 4.32 (2H, d, *J* = 5.9 Hz, NCH_2_), 4.33 (2H, d, *J* = 4.9 Hz, NCH_2_), 5.41 (1H, s, 4-CH), 7.15 (2H, d, *J* = 7.0 Hz), 7.20–7.40 (8H, m), 7.90 (2H, unres. t, NH). ^19^F NMR (471 MHz, CDCl_3_) *δ Z-***9d** (58%): 85.1 (s, CF_3_); *E-***9d** (42%): 84.9 (s, CF_3_). ^13^C NMR (126 MHz, CDCl_3_) *δ Z-***9d** (58%): 41.1 (2-CH_2_), 44.0 (NCH_2_), 48.0 (NCH_2_), 90.1 (4-CH), 117.3 (q, *J* = 288.4 Hz), 127.1 (2C Ph), 127.8 (2C Ph), 127.9 (C Ph), 128.2 (C Ph), 128.9 (2C Ph), 129.2 (2C Ph), 135.7 (C Ph), 137.3 (C Ph), 164.9 (1-CO), 165.3 (C-3), 177.0 (q, *J* = 33.4 Hz, 5-CO). *E-***9d** (42%): 40.5 (2-CH_2_), 43.5 (NCH_2_), 48.3 (NCH_2_), 86.3 (4-CH), 117.7 (q, *J* = 290.5 Hz, CF_3_), 127.2 (2C Ph), 127.4 (C Ph), 127.7 (2C Ph), 128.3 (C Ph), 128.6 (2C Ph), 129.1 (2C Ph), 134.8 (C Ph), 137.6 (C Ph), 163.5 (1-CO), 168.0 (C-3), 176.9 (q, *J* = 32.1 Hz, 5-CO). IR (ATR) ν 3308, 3062, 3033, 1641 (C=O), 1549, 1264,1165, 1124 (CF_3_). Anal. Calculated for C_20_H_19_F_3_N_2_O_2_**·**0.33H_2_O: C 62.82; H 5.18; N 7.33. Found: C 63.01; H 5.10; N 7.09.

*N-Butyl-3-(butylamino)-6,6,6-trifluoro-5-oxohex-3-enamide* (**9e**). The reaction was carried out for 72 h at 60 °C. Yield 12 mg (7%), large gray crystals, mp 79–80 °C. ^1^H NMR (500 MHz, CDCl_3_) *δ Z-***9e** (65%): 0.93 (3H, t, *J* = 7.2 Hz, CH_3_), 0.95 (3H, t, *J* = 7.2 Hz, CH_3_), 1.32 (quint, 2H, *J* = 7.2 Hz, NCH_2_CH_2_), 1.43 (2H, sex, *J* = 7.2 Hz, CH_2_CH_3_), 1.49 (2H, sex, *J* = 7.2 Hz, CH_2_CH_3_), 1.63 (2H, quint, *J* = 7.2 Hz, NCH_2_CH_2_), 3.27 (2H, s, 2-CH_2_), 3.28 (2H, q, *J* = 6.3 Hz, NCH_2_), 3.42 (2H, q, *J* = 6.6 Hz, NCH_2_), 5.36 (1H, s, 4-CH), 5.65 (1H, s, NH amide), 11.08 (1H, s, NH enamine); *E-***9e** (35%): 0.90 (3H, t, *J* = 7.2 Hz, CH_3_), 0.96 (3H, t, *J* = 7.2 Hz, CH_3_), 1.34 (quint, 2H, *J* = 7.2 Hz, NCH_2_CH_2_), 1.39–1.53 (4H, masked, CH_2_CH_3_), 1.65 (2H, quint, *J* = 7.2 Hz, NCH_2_CH_2_), 3.15–3.24 (4H, m, NCH_2_), 3.75 (2H, s, 2-CH_2_), 5.32 (1H, s, 4-CH), 7.10 (1H, s, NH amide), 7.36 (1H, s, NH enamine). ^19^F NMR (471 MHz, CDCl_3_) *δ Z-***9e** (65%): 85.04 (s, CF_3_); *E-***9e** (35%): 84.90 (s, CF_3_). IR (ATR) ν 3289, 2958, 2872, 1591 (C=O), 1548, 1306, 1184, 1114 (CF_3_). HRMS (ESI) *m/z* [M + H]^+^. Calculated for C_14_H_25_F_3_N_2_O_2_: 309.1790. Found: 309.1799.

### 3.7. Synthesis of Compound ***10***

*1,3,4,5-Tetrahydro-4-(3,3,3-trifluoro-2-oxopropylidene)-2H-1,5-benzodiazepin-2-one* (**10**). Benzene-1,2-diamine (63 mg, 0.58 mmol) was added to a solution of trifluorotriacetic acid (**2**) (100 mg, 0.56 mmol) in 1,4-dioxane (1 mL) with stirring. The reaction was carried out for 24 h at room temperature. Then, the reaction mixture was acidified with aqueous HCl (1 mL, 3M). The precipitate was filtered and washed with water. The crude product was recrystallized from hexane. Yield 51 mg (33%), fine white powder, mp 244–245 °C (decomp.; lit. mp 243–245 °C [43]). ^1^H NMR (500 MHz, CDCl_3_) *δ* 3.31 (2H, s, CH_2_), 5.68 (1H, s, CH), 7.12 (1H, dd, *J* = 7.6, 1.6 Hz, H Ar), 7.24–7.34 (3H, m, H Ar), 8.17 (1H, s, 1-NH), 12.57 (1H, s, 5-NH). ^19^F NMR (471 MHz, CDCl_3_) *δ* 84.8 (s, CF_3_). HRMS (ESI) m/z [M + H]^+^. Calculated for C_14_H_25_F_3_N_2_O_2_: 271.0694. Found: 271.0673. The analytical data are in consistence with the literature [43].

### 3.8. Synthesis of Compounds ***13*** and ***14***

*N’-Phenyl-2-(1-phenyl-5-(trifluoromethyl)-1H-pyrazol-3-yl)acetohydrazide* (**13**) and *N’-phenyl-2-(1-phenyl-3-(trifluoromethyl)-1H-pyrazol-5-yl)acetohydrazide* (**14**). Phenylhydrazine (126 mg, 1.17 mmol) was added to a solution of trifluorotriacetic acid (**2**) (100 mg, 0.56 mmol) in EtOH (1 mL) with stirring. The reaction was carried out for 24 h at room temperature. Then, the reaction mixture was acidified with aqueous HCl (1 mL, 3M). The precipitate was filtered and washed with water. The crude product was recrystallized from hexane. Yield 122 mg (61%), yellow powder, mp 136–138 °C. The mixture of isomers **13** and **14** (82:18) did not separate by column chromatography. IR (ATR) ν 3281, 1657 (C=O), 1603, 1504, 1356, 1083 (CF_3_). HRMS (ESI) m/z [M + H]^+^. Calculated for C_18_H_16_F_3_N_4_O: 361.1276. Found: 361.1276.

**13**: ^1^H NMR (500 MHz, DMSO*-d*_6_) *δ syn*-isomer (88%): 3.65 (2H, s, CH_2_), 6.67–6.71 (1H, m, H Ph), 6.73 (2H, d, *J* = 7.8 Hz, H Ph), 7.03 (1H, s, H pyraz.), 7.11 (2H, t, *J* = 7.8 Hz, H Ph), 7.49–7.52 (2H, m, H Ph), 7.56–7.62 (3H, m, H Ph), 7.60–8.03 (br. s, 1H, PhNH), 9.91 (1H, s, CONH); *anti*-isomer (12%): 3.71 (2H, s, CH_2_), 6.67–6.71 (2H, m, H Ph), 6.76 (1H, t, *J* = 7.3 Hz, H Ph), 6.95 (1H, s, H pyraz.), 7.18 (2H, t, *J* = 7.8 Hz, H Ph), 7.40–7.43 (2H, m, H Ph), 7.52–7.56 (3H, m, H Ph), 8.06 (1H, s, PhNH), 9.22 (1H, s, CONH). ^19^F NMR (471 MHz, DMSO*-d*_6_) *δ anti*-isomer (12%): 106.18 (s, CF_3_); *syn*-isomer (88%): 106.13 (s, CF_3_). ^13^C NMR (126 MHz, CDCl_3_) *δ syn*-isomer: 34.3 (CH_2_), 109.0 (C-4 pyraz.), 113.5 (2C Ph), 119.4 (q, *J* = 269.4 Hz, CF_3_), 121.3 (C Ph), 125.6 (2C Ph), 129.16 (2C Ph), 129.20 (2C Ph), 129.6 (C Ph), 133.9 (q, *J* = 39.7 Hz, C-5 pyraz.), 138.7 (C Ph), 146.2 (C Ph), 147.6 (C-3 pyraz.), 168.9 (CO); *anti*-isomer: most of the signals are masked due to low concentration.

**14**: ^1^H NMR (500 MHz, DMSO*-d*_6_) *δ syn*-isomer (71%): 3.78 (2H, s, CH_2_), 6.55 (2H, d, *J* = 7.6 Hz, H Ph), 6.69 (1H, t, *J* = 7.4 Hz, H Ph), 6.87 (1H, s, H pyraz.), 7.10 (2H, t, *J* = 7.5 Hz, H Ph), 7.56–7.62 (5H, m, H Ph), 7.73 (1H, s, PhNH), 9.83 (1H, s, CONH); *anti*-isomer (29%): 3.79 (2H, s, CH_2_), 6.56 (2H, d, *J* = 7.3 Hz, H Ph), 6.76 (1H, t, *J* = 7.3 Hz, H Ph), 6.83 (1H, s, H pyraz.), 7.14 (2H, t, *J* = 7.5 Hz, H Ph), 7.43–7.47 (2H, m, H Ph), 7.51–7.55 (3H, m, H Ph), 7.95 (1H, s, PhNH), 9.25 (1H, s, CONH). ^19^F NMR (471 MHz, DMSO*-d*_6_) *δ anti*-isomer (29%): 101.91 (s, CF_3_), *syn*-isomer (71%): 101.83 (s, CF_3_). The analytical data are in consistence with the literature [35].

## Data Availability

Data are contained within the article and Appendix A.

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
