# Peer review of "Reactions of Trifluorotriacetic Acid Lactone and Hexafluorodehydroacetic Acid with Amines: Synthesis of Trifluoromethylated 4-Pyridones and Aminoenones"

_molecules, 2022, doi:10.3390/molecules27207098_

Round 1
Reviewer 1 Report
The manuscript by V. Y. Sosnovskikh and co-workers is devoted to the study of the hexafluorodehydroacetic acid and trifluorotriacetic acid lactone reactivity in the reactions with amines. Indeed, despite the fact that the chemistry of non-fluorinated analogues is well investigated, there is no information under the studied pyranones. Strong acceptor trifluoromethyl and trifluoroacetyl groups significantly affect the molecules reaction centers and, as shown by the authors, are capable of radically changing the direction and selectivity of reactions. In addition, their presence in the structures of pyridones and diazepines synthesized from DHA-f6 with arylamines and TAL-f3 with ortho-phenylenediamine, respectively, increases the pharmacological potential of the molecules. A wide range of amines has been tested, covering aliphatic, aromatic, heteroaromatic derivatives, as well as diamines. Their activity in reactions with the studied objects was analyzed, the dependences of the reaction time and product yields on the nature of substituents in amines were derived. The authors proposed rational mechanisms of product formation.
Due to the high potential of practical significance, the results obtained by the authors deserve publication in the chosen journal. However, the manuscript is not without flaws that need to be addressed. And taking into account the remarks, the work can be accepted for publication in “Molecules”.

Reviewer 2 Report
The paper describes the synthesis of trifluoromethylated 4-Pyridones and Aminoenones from Trifluorotriacetic Acid Lactone and Hexafluorodehydroacetic Acid. The work is interesting and the presentation is good. The compounds prepared are novel but they are not fully characterised. This needs to be corrected for the manuscript to be accepted for publication.
Based on that I suggest that the paper is accepted after the following are addressed.
Corrections/comments:
- The use of a flame burner in the synthesis of compound 1 is rather odd. Why did the authors not try a different way of heating that could be better controled (like an oil bath)?
- Table 1: It is not clear how the end of the reaction was decided. Was there starting material left after the reactions were stopped at 24 h?
- Table 1: Perhaps a good idea would be to use an anhydrous Bronsted acid (ie. PTSA) or a Lewis acid to catalyse a possible imine formation. Also, addition of a dehydrating agent such as molecular sieves or Al2O3 could help drive the reaction to completion as loss of water is observed.
- Experimental: The characterisation data are not consistent with all compounds. Some compounds are missing the mass spec data (3a, 3c, 3e, B', 4, 9a. 9c,9d), some compounds are missing IR data (B') and some compounds are missing 13C NMR data (3b, 3j, 3k, B',4, 9c, 10). All these data needs to be provided.
- The supporting information file needs to be improved. Each spectrum needs to have the aquisition information visible and the whole spectrum must be visible (15-0 ppm for 1H and 220-0 ppm for 13C NMR). Moreover, please provide zoomed images for all regions where the peaks are not clearly visible.
Round 2
Reviewer 1 Report
I am satisfied with the work authors done.
Nevertheless the phrase “Thus, aniline derivatives bearing donor substituents (4-MeO, 4-F)…” is misleading. It is necessary to give an explanation, which fluorophenyl derivative is meant.
The manuscript may be accepted for publication.
Author Response
Dear Editorial Board,
We would like to provide the following response for the second round of review.
Reviewer 1.
We thank the reviewer 1 for all corrections and comments on our manuscript and we are glad that our efforts have satisfied him/her. We have also put some corrections to the text at the line 95 to improve the clarity of the sentence about an influence of a substituent on the reactivity of anilines towards DHA-f6 (“Thus, aniline derivatives bearing π-donor substituents (MeO, F) at the 4-position react considerably faster whereas those bearing strong acceptor substituents (Ac, CF3) at the 3-position or a weak acceptor substituent (Br) at the 4-position react at comparable rate but give slightly lower yields”).
Reviewer 2 Report
Response to the authors:
Elemental analysis and HRMS are usually considered as alternative ways to prove the composition of a compound that’s why we provided only one of them. The missing spectra were added where possible.
I disagree with that. Mass spectrometry, whether high resolution or low resolution, is not "optional". It is necessary to prove that you have the correct compound. Lack of complete characterisation is an important reason for the rejection of this manuscript.
We do not feel the acquisition information as well as empty downfield regions to be of significant importance for readers. Zoomed images would rather overwhelm spectra in our opinion and are rarely included in supporting information files. All multiplets and chemical shifts are described in experimental section.
I again disagree. Not showing a part of the spectrum shows that you are trying to hide something. It is of the interest to the readers to see that your products are clean.
Author Response
Dear Editorial Board,
We would like to provide the following response for the second round of review.
Reviewer 2.
We thank the reviewer 2 for his/her thoroughness and efforts to improve our article. We respect high scientific demands and traditions of other research groups. But we also think that a characterization of a compound is complete when no doubts on its structure are left. We have provided enough data to prove the structure of our products on our opinion, and if the reviewer 2 has some thoughts on alternative ones we would be thankful to him/her for sharing them. Preparing our manuscript, we just followed the instructions for a characterization of new compounds recommended by the Molecules Editorial Board which refer to the ACS rules. We have chosen the JOC rules (https://publish.acs.org/publish/author_guidelines?coden=joceah#supporting_information) as our article is a synthetic one and here are some quotations from them:
“Data Requirements
… Identity. Evidence for documenting the identity of new compounds should include both proton and carbon NMR data and either MS accurate mass (HRMS) or elemental analysis data.
… NMR. Proton and carbon NMR resonances should be listed for each new compound, with the normal full range of chemical shifts displayed (usually 10–0 ppm for proton; 200–0 ppm for carbon); the solvent and instrument frequency should be identified.”
It should be also noted that the aminoenones occurred to be incompatible with electrospray ionization used in our spectrometer due to facile intramolecular cyclisation with the loss of amine.
